# Forefoot Deformities in Patients with Rheumatoid Arthritis: Mid- to Long-Term Result of Joint-Preserving Surgery in Comparison with Resection Arthroplasty

**DOI:** 10.3390/ijerph182111257

**Published:** 2021-10-26

**Authors:** Yuya Takakubo, Yoshihiro Wanezaki, Hiroharu Oki, Yasushi Naganuma, Junichiro Shibuya, Ryusuke Honma, Akemi Suzuki, Hiroshi Satake, Michiaki Takagi

**Affiliations:** 1Department of Orthopaedic Surgery, Faculty of Medicine, Yamagata University, Yamagata 990-9585, Japan; wane3775@yahoo.co.jp (Y.W.); j98c0547@yahoo.co.jp (H.O.); yasushi804@msn.com (Y.N.); jun820702@gmail.com (J.S.); ryuhonma32@gmail.com (R.H.); akemis@med.id.yamagata-u.ac.jp (A.S.); hsatake@med.id.yamagata-u.ac.jp (H.S.); mtakagi@ameria.org (M.T.); 2Department of Rehabilitation, Faculty of Medicine, Yamagata University, Yamagata 990-9585, Japan; 3Faculty of Orthopaedic Surgery, Yamagata Saisei Hospital, Yamagata 990-9585, Japan; 4Faculty of Orthopaedic Surgery, Yamagata Prefectural Central Hospital, Yamagata 990-2292, Japan; 5Department of Orthopaedic Surgery, Izumi Orthopaedic Hospital, Sendai-shi 981-3121, Japan

**Keywords:** foot, joint-preserving surgery, rheumatoid arthritis

## Abstract

Background: Joint-preserving surgery for the forefoot has been increasingly performed for rheumatoid arthritis (RA). We compared joint-preserving surgeries with resection arthroplasty for RA in the forefoot. Methods: Forefoot surgeries were performed on 62 toes in 42 patients with RA (men: 2; women: 40) between 2002 and 2018. Three groups were compared: PP—31 toes treated with joint-preserving surgery involving the modified Mann method for the big toe and offset osteotomy for lesser toes, PR—15 toes treated with joint-preserving surgery for the big toe and resection arthroplasty for lesser toes, and RR—16 toes treated with resection arthroplasty for all the toes. Results: The PP group had significantly higher mean scores on a scale for RA in the foot and ankle at the latest follow-up than the RR group (86 vs. 75 points; *p* < 0.05). Hallux valgus (angle > 20°) of the big toe at the latest follow-up recurred in 10 (32%), 9 (60%), and 16 (100%) patients in the PP, PR, and RR groups, respectively. A revision surgery was performed in one patient each in the PP and PR groups. Conclusions: Joint-preserving surgery is superior to resection arthroplasty in preventing function loss and the recurrence of hallux valgus.

## 1. Introduction

Forefoot deformities have been reported to occur in approximately 90% of the patients with rheumatoid arthritis (RA) [1,2]. Forefoot surgery for severe foot deformity due to RA has already been performed in 25–40% of such patients [3,4,5].

Various surgical procedures have been reported for forefoot deformity [2,5,6,7]. Representative surgical procedures include metatarsophalangeal (MTP) joint-preserving surgery, resection arthroplasty, arthrodesis, and artificial arthroplasty of MTP joint deformity for hallux valgus deformity and deformity of lesser toes.

Resection arthroplasty and arthrodesis of the MTP joint have been the standard interventions for foot deformity due to RA [1,4,5]. Biologic treatment with methotrexate has been used in Japan since 2003 to prevent inflammatory joint destruction in RA [8,9]. After a paradigm shift in the approach to RA therapy, joint-preserving surgeries for RA forefoot deformity have been increasingly performed, particularly in Japan [5,6,7,10,11,12,13,14,15,16,17]. However, 10–20% patients who even received biologics showed no or minor improvements such as difficult-to-treat RA [18]. Particularly, forefoot deformity occurred in most patients with RA. Therefore, the surgery for RA forefoot deformity may be needed at a constant rate, although the use of biologics may slow the rate of foot joint erosion [19].

Thus, in this study, we aimed to compare the mid- to long-term results of MTP joint-preserving surgeries involving the modified Mann method for the big toe and offset osteotomy for lesser toes with those of resection arthroplasty.

## 2. Patients and Methods

### 2.1. Patients and Surgical Procedures

All the patients with RA included in this study satisfied the criteria for RA established in 1987 by the American College of Rheumatology [20] or in 2010 by the American College of Rheumatology/European League against Rheumatism [21]. Forefoot surgeries were performed on 62 toes in 42 patients with RA from 2002 to 2018. Of these toes, 31 (in 22 patients) were treated with joint-preserving surgery involving the modified Mann method for the big toe and offset osteotomy for lesser toes (PP group), 15 toes (in nine patients) were treated with joint-preserving surgery involving the modified Mann method for the big toe and resection arthroplasty for lesser toes (PR group), and 16 toes (in 11 patients) were treated with resection arthroplasty for the big and lesser toes (RR group).

A total of 42 patients (40 women and 2 men) were treated by five surgeons (Y.T., H.O., Y.N., J.S., and M.T.). The surgeons performed resection arthroplasty if severe subluxation and dislocation of the MTP joint with contractures and stiffness were observed upon manipulation or stress radiographs.

The mean age of the patients at the time of surgery was 61 years (range, 35–76). The mean duration of RA was 13.3 years (range, 4–18). According to the Steinbrocker functional classification, RA was stage III in 24 patients, stage IV in 18, class 2 in 27, and class 3 in 15. The mean level of C-reactive protein (CRP) at the time of surgery was 1.6 mg/dL (range, 0.1–5.3), and the mean Disease Activity Score in 28 Joints (DAS28) calculated with CRP (DAS28-CRP) was 3.7 (range, 1.9–5.2). Of the 42 patients, 10 (24%) received biologics (three received infliximab, six—etanercept, and one—abatacept), 23 (55%) received methotrexate (mean dosage: 6.2 mg/week), and 12 (29%) received prednisolone (mean dosage: 3.4 mg/day; Table 1).

The study protocol was approved by the Ethics Committee of Yamagata University, Japan (No. H27-162). All the procedures were performed in accordance with the relevant guidelines and regulations. This study was conducted in accordance with the Declaration of Helsinki.

### 2.2. Analyses

We rated the clinical outcomes according to the Japanese Society for Surgery of the Foot (JSSF) RA foot and ankle scale [22]. Radiographs were taken to measure the hallux valgus angle (HVA), the M1M2 angle, and the M1M5 angle during the mean follow-up period of 9.4 years (range, 3–16). We defined recurrence as a new deformity in which the HVA was > 20° and severe recurrence as a new deformity in which the HVA was > 40°.

### 2.3. Statistical Analysis

To perform one-factor analysis of variance, the Mann–Whitney U test of the null hypothesis, and the Tukey–Kramer tests, we used the PASW 25 software (SPSS Institute Inc., Chicago, IL, USA). We considered *p*-values < 0.05 as significant.

### 2.4. Surgical Procedure

#### 2.4.1. MTP Joint-Preserving Surgery with the Modified Mann Method for the Big Toe and Offset Osteotomy for Lesser Toes

For the PP group, the modified Mann method was used in combination with (a) elevation of the abductor hallucis tendon, which was dislocated to the plantar side of the big toe toward the tibial side of the metatarsal bone; (b) cutting of the MTP joint capsule of the big toe on the tibial side in a Y-shape, which was plicated and shortened for the closing of the joint capsule after the modified Mann osteotomy; (c) partial resection of the distal end of the adductor hallucis tendon and transverse metatarsal ligament from the lateral aspect of the fibular sesamoid and base of the proximal phalanx; (d) removal of the connective tissue that was inflamed as a result of bunions; (e) proximal osteotomy of the metatarsal bone for the shortening and correction of the varus deformity and its internal rotational deformity with the use of a fixed 1.6-mm Kirschner wire; and (f) offset osteotomy of the metatarsal bone with the MTP joints of the lesser toes [6]. An example is shown in Figure 1.

#### 2.4.2. MTP Joint-Preserving Surgery Involving the Modified Mann Method for the Big Toe and Resection Arthroplasty for Lesser Toes

For the PR group, the modified Mann method was used for the big toe following the same procedure as that used for the PP group. Resection arthroplasty of the MTP joint was performed for the lesser toes affected by severe subluxation, dislocation of these joints, contracture, and stiffness. The resection included 10–15 mm of the end of the metatarsal head, followed by fixation using a Kirschner wire from the tip of the toe to the tarsometatarsal joint through the MTP joint for 3 weeks. An example is shown in Figure 2.

#### 2.4.3. Resection Arthroplasty for the Big and Lesser Toes

Resection arthroplasty for the big and lesser toes was performed in the same way as that described for the PR group, and soft tissue repair was performed as that described for the PP group. An example is shown in Figure 3.

### 2.5. Postoperative Protocol

The patients were able to walk 1 or 2 days after surgery with a non-weight-bearing orthosis of the forefoot [6]. After 3 weeks, the wires used for the temporary fixation of the MTP joints were removed and the patients were allowed to walk with full-weight-bearing shoes equipped with arch support. After discharge from the hospital, the patients wore the shoes with arch support and performed range-of-motion (ROM) exercises with their feet by themselves every day.

## 3. Results

### 3.1. JSSF Scores

The mean JSSF scores were 38.7 (range, 21–57) before surgery and 78.6 (range, 50–98) at the latest follow-up. The mean JSSF score at the latest follow-up was significantly higher in the PP group than in the RR group (*p* < 0.05; Table 2).

### 3.2. Radiographic Measurements

The mean HVA angles were 41.7° (range, 17–67) before surgery and 23.8° (range, 4–60) at the latest follow-up. The mean HVAs at the latest follow-up did not differ significantly between the three groups (Table 3). Hallux valgus (>20°) at the latest follow-up recurred in 10 patients (32%) in the PP group, 9 (60%)—in the PR group, and all 16 (100%)—in the RR group. Severe hallux valgus (>40°) at the latest follow-up recurred in one patient each in the PP (3.2%), PR (6.6%), and RR (6.3%) groups. The mean M1M2 angles were 14.9° (range, 8–22) before surgery and 11.9° (range, 1–27) at the latest follow-up. The mean M1M2 angle at the latest follow-up was significantly smaller in the PP group than in the RR group (*p* < 0.05; Table 4). The mean M1M5 angle was 33.5° (range, 18–48) before surgery and 28.1° (range, 19–40) at the latest follow-up. The mean M1M5 angle at the latest follow-up was significantly smaller in the PP and PR groups than in the RR group (*p* < 0.05; Table 5).

### 3.3. Complications of Surgery

Two revision surgeries were performed—one in the PP group 3 years after the first surgery and the other in the PR group 8 years in the PR group after the first surgery—because of the recurrence of painful plantar callosities in lesser toes. Delayed healing of the surgical wound was observed in one, one, and three patients in the PP, PR, and RR groups, respectively. Nonunion was not observed in this study.

## 4. Discussion

Joint-preserving surgery has been increasingly performed for forefoot deformities in patients with RA [5]. Recently, therapies for RA, including biologics and Janus kinase (JAK) inhibitors, have been developed to prevent inflammatory joint destruction [8]. As a result, function loss can be prevented, the ROM can be extended, and the alignment of the smooth arch of the foot and the ability to push off are improved [1,15,23]. However, 10–20% of the patients with RA do not respond to biologics or JAK inhibitors or show only minor improvements [9]. The discussion about the indications for MTP joint-preserving surgery is thus ongoing [5,6].

In this study, joint-preserving forefoot surgery involving the modified Mann method for the big toe and offset osteotomy for the lesser toes (PP group) produced results superior, according to the JSSF scale scores and radiographic measurements, to those observed in the PR and RR groups. However, the recurrence rate of the hallux valgus deformity of the big toe in this study was higher (26%) than that reported in other studies regarding MTP joint-preserving surgery [5,14,24]. One of the reasons for that was the difference in indications for joint-preserving surgery. Niki et al. [14] reported excellent results with no recurrence of the hallux valgus deformity after combination metatarsal osteotomies for shortening, including arthrodesis of the tarsometatarsal joint of the big and second toes to treat forefoot deformities due to RA. The mean postoperative follow-up duration in their study was 76.6 months (range, 64–108). They included patients with RA in clinical remission, which was defined by the mean DAS28-CRP of 1.65 (range, 1.13–2.23), and excluded those with severe destruction (Larsen’s grade 4 or 5) of the metatarsal head [14].

Fukushi et al. [24] also reported excellent results with no recurrence of hallux valgus after biplane osteotomy for the big toe and Weil shortening osteotomy for lesser toes involved in forefoot deformities due to RA. The mean postoperative follow-up duration in their study was 28 months (the range was not mentioned). Although the DAS28-CRP scores were not described, all their patients were in remission or had low disease activity. Yano et al. [5,15] included Larsen’s grade 4 severity in the indications for proximal rotational closing wedge osteotomy of the big toe and modified shortening oblique osteotomy of the lesser toes; however, hallux valgus recurred in 11 (10.5%) of the 105 feet studied. In their study, the mean duration of follow-up for the entire cohort was 6.0 years (standard deviation, 0.9; range, 5.0–7.4). The mean DAS28 calculated with erythrocyte sedimentation rate was 3.1 (range, 2.5–4.3) at the time of surgery in their study.

In our study, the mean DAS28-CRP for the PP group was 3.8 (range, 1.9–5.6), and the indication for joint-preserving surgery was Larsen’s grade 4 severity. Our cohort included patients with challenging cases involving high disease activity and severe joint destruction, particularly those who underwent joint-preserving surgery in or before 2003. In Japan, biologics have been approved for the treatment of RA since 2003 and the methotrexate dosage had been restricted to < 8 mg/week until 2011 [8]. The patients who underwent treatment later in this study exhibited improved disease activity (data not shown), and the indications for joint-preserving surgery were more limited than at the beginning of this study. Currently, surgeons try to keep disease activity low before and after surgery to prevent the progression of bone destruction; partial arthrodesis of the tarsometatarsal joint, such as combination metatarsal osteotomies for shortening [7,15], or additional osteotomy of the distal phalanx, such as Akin osteotomy, are performed for severe deformities [5].

In our study, the outcomes in the PR group were superior to those in the RR group but inferior to those in the PP group. Hulse et al. reported that a subsequent resection of the first metatarsal head was necessary at the second surgery because of a high rate of pain recurrence in the patients receiving MTP joint arthroplasty with metatarsal head resections of lesser toes and without that of the big toe at the initial surgery [25]. In this study, the PR group exhibited the recurrence of the hallux valgus deformity of the big toe at a high rate (60%) at the latest follow-up; however, severe deformity recurred in only one patient. The RR group exhibited less correction of the HVA, the M1M2 angle, and the M1M5 angle than the other groups. In fact, all the patients in the RR group suffered from recurrence of the hallux valgus deformity by the latest follow-up.

Two patients in this study underwent revision surgery for the recurrence of painful plantar callosities of lesser toes. The recurrences resulted from the undercorrection of the deformity and insufficient dissection of adhesions around the metatarsal head of lesser toes [1,26].

Yano et al. [27] reported that in patients with RA, the results of a multiple regression analysis revealed longer surgery duration as a risk factor associated with delayed wound healing after forefoot surgeries such as joint-preserving surgery and resection arthroplasty. In our study, five patients demonstrated delayed surgical wound healing, although the surgery durations for them were similar to those in other patients (data not shown); however, no infection occurred. Delayed wound healing was more common in the RR (18.8%) group than in the PP (6.7%) and PR (3.2%) groups. The patients in the RR group were, on average, older than those in the other groups and showed a tendency to have higher disease activity.

Perioperative rehabilitation is important for achieving better outcomes after surgery. Early partial weight bearing prevents the disuse-related atrophy of muscles and bones [1]. Hirao et al. reported that after 6 months from surgery, the patients who started exercising early (2 weeks after surgery) had a better ROM of lesser toes than those who did not [28]. In our study, the temporary wires used for the fixation of the MTP joints were removed 3 weeks after surgery and the patients then started ROM exercises of lesser toes. The time of temporary wire removal and ROM exercise initiation should be reconsidered for better long-term outcomes.

This study has several limitations. Firstly, the patient-reported outcome measure was not evaluated. A self-administered foot evaluation questionnaire (SAFE-Q) was developed in 2013 for Japanese patients for foot evaluation [22]. Ebina et al. reported that SAFE-Q scores were significantly better after joint-preserving surgery than after resection arthroplasty [29]. Secondly, our study population was relatively small. Thirdly, this study was retrospective in nature. Fourthly, the surgical procedure depended on the surgeon’s discretion, which might have led to selection bias. In the future, large randomized prospective studies with a multicenter approach and a longer follow-up duration should be conducted.

## 5. Conclusions

Joint-preserving forefoot surgery involving the modified Mann method for the big toe and offset osteotomy for lesser toes may prevent function loss after the restoration of forefoot deformities compared with resection arthroplasty for the big and lesser toes in combination with biologic therapy for RA.

## Figures and Tables

**Figure 1 ijerph-18-11257-f001:**
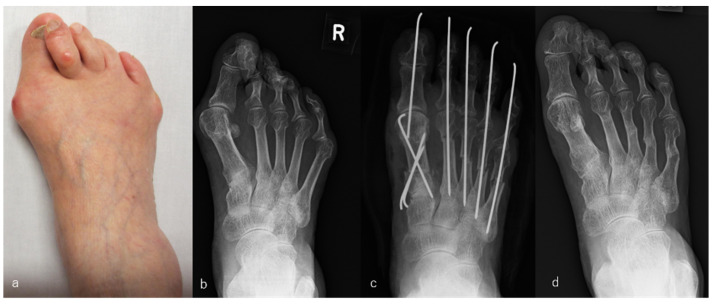
Rheumatoid arthritis (RA) in a 78-year-old woman. Metatarsophalangeal joint-preserving surgery was performed involving the modified Mann method for the big toe and offset osteotomy for lesser toes. RA had been present for 6 years. The patient received bucillamine (200 mg/day) and sulfasalazine (1000 mg/day). Macrographic and radiographic views of her right foot before surgery (**a**,**b**), immediately after surgery (**c**), and 7 years after surgery, which was the latest follow-up (**d**). The score on the Japanese Society for Surgery of the Foot RA foot and ankle scale improved from 40 points before surgery to 92 points at the 7-year follow-up. The mean hallux valgus angle improved from 45° before surgery to 14° immediately after surgery and was 19° at the 7-year follow-up. The patient was satisfied with the clinical results at the 7-year follow-up, although the correction of the hallux valgus angle had decreased slightly compared with that immediately after surgery.

**Figure 2 ijerph-18-11257-f002:**
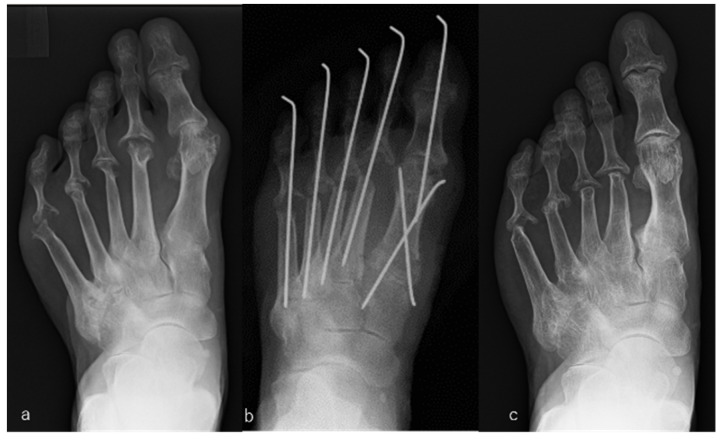
Rheumatoid arthritis (RA) in a 53-year-old woman. Metatarsophalangeal joint-preserving surgery involving the modified Mann method was performed for the big toe, and resection arthroplasty was performed for lesser toes. RA had been present for 29 years. The patient received methotrexate (4 mg/week), tacrolimus (2 mg/day), and sulfasalazine (1000 mg/day). Radiographic views of her left foot before surgery (**a**), immediately after surgery (**b**), and 8 years after surgery, which was the latest follow-up (**c**). Scores on the Japanese Society for Surgery of the Foot RA foot and ankle scale improved from 50 points before surgery to 84 points at the 8-year follow-up. The mean hallux valgus angle improved from 32° before surgery to 9° immediately after surgery and was 16° at the 8-year follow-up. The patient was satisfied with the clinical results at the 8-year follow-up, although the correction of the hallux valgus angle had decreased slightly and the metatarsal bone had shortened compared with the values immediately after surgery.

**Figure 3 ijerph-18-11257-f003:**
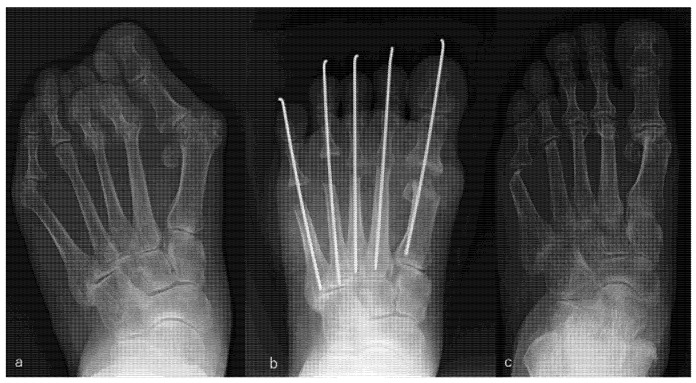
Rheumatoid arthritis (RA) in a 56-year-old woman. Metatarsophalangeal joint resection arthroplasty for the big and lesser toes was performed. RA had been present for 26 years. The patient received etanercept (50 mg/week), methotrexate (8 mg/week), and prednisolone (3 mg/day). Radiographic views of her left foot before surgery (**a**), immediately after surgery (**b**), and 9 years after surgery, which was the latest follow-up (**c**). Scores on the Japanese Society for Surgery of the Foot RA foot and ankle scale improved from 38 points before surgery to 79 points at the 9-year follow-up. The mean hallux valgus angle improved from 48° before surgery to 4° immediately after surgery and was 23° at the 9-year follow-up. The patient was satisfied with the clinical results at the 9-year follow-up, although the correction of the hallux valgus angle had decreased slightly and the metatarsal bone had shortened compared with the values immediately after surgery.

**Table 1 ijerph-18-11257-t001:** Characteristics of patients with RA.

Characteristic		PP Group	PR Group	RR Group
No. of feet/cases		31/22	15/9	16/11
Age (years)	Mean	55	60	66
	Range	(35–71)	(51–76)	(42–74)
Male/female		0/22	1/8	1/10
Duration of RA	Mean	12.1	16	11
	Range	5–18	4–24	6–2
Steinbrocker stage		III (*n* = 15), IV (*n* = 7)	III (*n* = 4), IV (*n* = 5)	III (*n* = 5), IV (*n* = 6)
Steinbrocker class		2 (*n* = 14), 3 (*n* = 8)	2 (*n* = 6), 3 (*n* = 3)	2 (*n* = 7), 3 (*n* = 4)
DAS28-CRP	Mean	3.8	3.6	3.9
	Range	1.9–5.2	2.6–5.2	3.7–5.0
CRP level	Mean	1.5	1.3	3.3
	Range	0.1–5.3	0.1–4.5	0.1–4.3
No. of patients receiving biologic treatment, *n* (%)		5	3	2
		(22.7)	(33.3)	(18.2)
No. of patients receiving MTX, *n* (%)		13	6	4
		(59.1)	(66.7)	(36.4)
Mean dose of MTX		6 mg/week	5 mg/week	3 mg/week
No. of patients receiving PSL, *n* (%)		4	3	5
		(18.2)	(33.3)	(45.5)
Mean dose of PSL,		2 mg/day	2 mg/day	4 mg/day
Duration of follow-up (years)	Mean	8.4	7.2	9
	Range	3–13	3–15	6–12

DAS28-CRP, Disease Activity Score in 28 Joints calculated with CRP; CRP, C-reactive protein; MTX, methotrexate; PSL, prednisolone. The PP group underwent joint-preserving surgery involving the modified Mann method for the big toe and offset osteotomy for lesser toes. The PR group underwent joint-preserving surgery involving the modified Mann method for the big toe and resection arthroplasty for lesser toes. The RR group underwent resection arthroplasty for the big and lesser toes.

**Table 2 ijerph-18-11257-t002:** Mean JSSF RA foot and ankle scale in three groups.

Group		Before Surgery	After Surgery	Latest Follow-Up
PP	Mean	45	86	79 *
	Range	21–55	72–98	61–98
PR	Mean	37	80	74
	Range	21–57	63–94	59–90
RR	Mean	36	75	69
	Range	23–45	68–83	61–81

JSSF, Japanese Society for Surgery of the Foot. RA, rheumatoid arthritis. The PP group underwent joint-preserving surgery involving the modified Mann method for the big toe and offset osteotomy for lesser toes. The PR group underwent joint-preserving surgery with the modified Mann method for the big toe and resection arthroplasty for lesser toes. The RR group underwent resection arthroplasty for the big and lesser toes. * Mean JSSF RA foot ankle scale at the latest follow-up in the PP group was significantly higher compared to the RR group (*p* < 0.05).

**Table 3 ijerph-18-11257-t003:** Mean HVA in the three groups.

Group		Before Surgery	After Surgery	Latest Follow-Up	Recurrence of Hallux Valgus (>20°) at the Latest Follow-Up
PP	Mean	41.4	12.1	21.6	*n* = 10/31 (32%)
	Range	17–67	9–25	4–60	
PR	Mean	38.3	9.9	20.1	*n* = 9/15 (60%)
	Range	21–63	0–12	9–49	
RR	Mean	46.3	15.0	27.9	*n* = 16/16 (100%)
	Range	30–67	4–26	22–40	

The PP group underwent joint-preserving surgery involving the modified Mann method for the big toe and offset osteotomy for lesser toes. The PR group underwent joint-preserving surgery involving the modified Mann method for the big toe and resection arthroplasty for lesser toes. The RR group underwent resection arthroplasty for the big and lesser toes.

**Table 4 ijerph-18-11257-t004:** Mean M1M2 angle in the three groups.

Group		Before Surgery	After Surgery	Latest Follow-Up
PP	Mean	14.6	9.2	9.6 *
	Range	8–21	1–16	1–16
PR	Mean	15.5	11.8	12.0
	Range	9–20	6–18	1–22
RR	Mean	15.7	12.0	17.6
	Range	12–22	6–20	6–27

The PP group underwent joint-preserving surgery involving the modified Mann method for the big toe and offset osteotomy for lesser toes. The PR group underwent joint-preserving surgery involving the modified Mann method for the big toe and resection arthroplasty for lesser toes. The RR group underwent resection arthroplasty for the big and lesser toes. * The mean M1M2 angle at the latest follow-up in the PP group was significantly smaller than that in the RR group (*p* < 0.05).

**Table 5 ijerph-18-11257-t005:** Mean M1M5 angle in the three groups.

Group		Before Surgery	After Surgery	Latest Follow-Up
PP	Mean	32.4	25.0	26.6 *
	Range	25–45	15–33	19–37
PR	Mean	33.8	24.9	26.6 *
	Range	18–48	15–31	18–40
RR	Mean	35.6	26.6	33.5
	Range	26–44	20–38	25–40

The PP group underwent joint-preserving surgery involving the modified Mann method for the big toe and offset osteotomy for lesser toes. The PR group underwent joint-preserving surgery involving the modified Mann method for the big toe and resection arthroplasty for lesser toes. The RR group underwent resection arthroplasty for the big and lesser toes. * The mean M1M5 angle at the latest follow-up was significantly smaller in the PP* and PR* groups than in the RR group (*p* < 0.05).

## Data Availability

All the data are available through cited publications.

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
