# Peer review of "Forefoot Deformities in Patients with Rheumatoid Arthritis: Mid- to Long-Term Result of Joint-Preserving Surgery in Comparison with Resection Arthroplasty"

_ijerph, 2021, doi:10.3390/ijerph182111257_

Round 1
Reviewer 1 Report
Is there any information about the reason of choosing different operation methods in these patient? Did these factors vary with operating surgeon, disease severity... which might confound the results?
Author Response
Responses to Reviewers’ Comments
Reviewer 1
Is there any information about the reason of choosing different operation methods in these patient? Did these factors vary with operating surgeon, disease severity... which might confound the results?
Response: We greatly appreciate you for highlighting these critical points.
We have added the following sentences in the Materials and Methods section of the revised manuscript:
“A total of 42 patients (40 women and 2 men) were treated by five surgeons (Y.T., H.O., Y.N., J.S., and M.T.). The surgeons performed resection arthroplasty if severe subluxation and dislocation of the MTP joint with contractures and stiffness were observed upon manipulation or on stress radiographs.”
We agree with you on the fact that the selection of different surgical methods in this study might have led to a selection bias with multiple factors. The final decision regarding the surgical procedures was made by the surgeon. We have added this as a limitation to our study.
“Fourth, the surgical procedure depended on the surgeon’s discretion, which might have led to a selection bias.”

Reviewer 2 Report
In the manuscript, Takakubo et al mainly investigated the the effect of joint preserving surgery and resection arthroplasty on forefoot deformities of RA patients and conducted the comparison between them. The study is well-designed, and sufficiently described in a logical way. I only have minor comments for the study.
(1) For the statistical analysis, the p value and symbol should be clarified in the table and be consistent with the context in the legend in Table 5, p<0.05 or p<0.01.
(2)The English language and grammar should be polished by a native speaker.
Author Response
Reviewer 2
In the manuscript, Takakubo et al mainly investigated the effect of joint preserving surgery and resection arthroplasty on forefoot deformities of RA patients and conducted the comparison between them. The study is well-designed, and sufficiently described in a logical way. I only have minor comments for the study.
Response: We truly appreciate the important points.
- For the statistical analysis, the p value and symbol should be clarified in the table and be consistent with the context in the legend in Table 5, p<0.05 or p<0.01.
Response: We apologize for the inconsistency. We have revised it to p<0.05* in Table 5.
(2)The English language and grammar should be polished by a native speaker.
Response: We apologize for the poor usage of English language and grammar in our previous manuscript. The English Editing and Manuscript Proofreading company Enago (www.enago.jp) has rechecked our revised manuscript.

Reviewer 3 Report
Thank you for this helpful contribution. The authors present a study on Forefoot deformities in patients with rheumatoid arthritis: Mid- to long-term result of joint-preserving surgery in comparison with resection arthroplasty. I applaud the effort of promoting these studies.
I want to thank you for the opportunity to review this manuscript. Joint-preserving surgery has been increasingly performed for forefoot deformity in patients with RA. The authors have presented sufficient data, it is recommended that this study be continued and provide data corresponding to a larger number of target patients to corroborate the findings of this study.
The authors have used appropriate statistical methods and the tables and figures have been correctly presented in the article. The conclusions are consistent with the evidence presented and are well reasoned. This is a well written, easy to read and well structured manuscript. The author should review the table presentation format. I write some comments below that could benefit the article:
The first section, the introduction could be expanded. It would be helpful to know what the biologic treatment used in the literature. In other words, use the published literature and your own thought process to link specific foot problems to biologic treatment. I suggest to include the following references:
-Nagy G, et al. EULAR definition of difficult-to-treat rheumatoid arthritis. Ann Rheum Dis. 2021 Jan;80(1):31-35. doi: 10.1136/annrheumdis-2020-217344. Epub 2020 Oct 1. PMID: 33004335; PMCID: PMC7788062.
- Ramos-Petersen L, et al. A Systematic Review to Identify the Effects of Biologics in the Feet of Patients with Rheumatoid Arthritis. Medicina (Kaunas). 2020 Dec 29;57(1):23. doi: 10.3390/medicina57010023. PMID: 33383830; PMCID: PMC7824565.
Is necessary a sub-section of :… “3.3. Complications of surgery…”
Conclusion. Conclusion must respond to the main aim of the study. It is advisable that conclusion be clear and concise. Therefore, it is not recommended that its length be greater than some lines. Limitations: broader studies are required.
References. I would like to congratulate you because 100% of the information you have referenced comes from articles published in scientific journals.
Congratulations on this wonderful article!! I think the authors have worked very closely considering methodology, outcomes. This will impact on international practice and help us reflect on the work we do. Thank you for this invitation to provide a review to evaluate this article for publication. I am delighted to have been invited to review this work.
Author Response
Reviewer 3
Thank you for this helpful contribution. The authors present a study on Forefoot deformities in patients with rheumatoid arthritis: Mid- to long-term result of joint-preserving surgery in comparison with resection arthroplasty. I applaud the effort of promoting these studies.
I want to thank you for the opportunity to review this manuscript. Joint-preserving surgery has been increasingly performed for forefoot deformity in patients with RA. The authors have presented sufficient data, it is recommended that this study be continued and provide data corresponding to a larger number of target patients to corroborate the findings of this study.
The authors have used appropriate statistical methods and the tables and figures have been correctly presented in the article. The conclusions are consistent with the evidence presented and are well reasoned. This is a well written, easy to read and well structured manuscript.
- The author should review the table presentation format. I write some comments below that could benefit the article:
Response: Thank you for the detailed feedback.
The conversion of Word to PDF resulted in the presentation issues. We have carefully reviewed the table and converted it to PDF again.
- The first section, the introduction could be expanded. It would be helpful to know what the biologic treatment used in the literature. In other words, use the published literature and your own thought process to link specific foot problems to biologic treatment. I suggest to include the following references:
-Nagy G, et al. EULAR definition of difficult-to-treat rheumatoid arthritis. Ann Rheum Dis. 2021 Jan;80(1):31-35. doi: 10.1136/annrheumdis-2020-217344. Epub 2020 Oct 1. PMID: 33004335; PMCID: PMC7788062.
- Ramos-Petersen L, et al. A Systematic Review to Identify the Effects of Biologics in the Feet of Patients with Rheumatoid Arthritis. Medicina (Kaunas). 2020 Dec 29;57(1):23. doi: 10.3390/medicina57010023. PMID: 33383830; PMCID: PMC7824565.
Response: Thank you for such an insightful suggestion. We have revised and expanded the Introduction section using the abovementioned references. We have added the following sentences in the Introduction and References sections of our revised manuscript:
“However, 10%–20% patients who even received biologics showed no or minor improvements such as difficult-to-treat RA [18]. Particularly, forefoot deformity occurred in most patients with RA. Therefore, the surgery for RA forefoot deformity may be needed at a constant rate, although the use of biologics may slow the rate of foot joint erosion [19].”
- Nagy, G.; Roodenrijs, N.M.T.; Welsing, PM.; Kedves, M.; Hamar, A.; van der Goes, M.C.; Kent, A.; Bakkers, M.; Blaas, E.; Senolt, L.; Szekanecz, Z.; Choy, E.; Dougados, M.; Jacobs, J.W.; Geenen, R.; Bijlsma, H.W.; Zink, A.; Aletaha, D.; Schoneveld, L.; van Riel, P.; Gutermann, L.; Prior, Y.; Nikiphorou, E.; Ferraccioli, G.; Schett, G.; Hyrich, KL.; Mueller-Ladner, U.; Buch, M.H.; McInnes, I.B.; van der Heijde, D.; van Laar, J.M. EULAR definition of difficult-to-treat rheumatoid arthritis. Ann. Rheum. Dis. 2021, 80, 31–35. doi: 10.1136/annrheumdis-2020-217344.
- Ramos-Petersen, L.; Nester, CJ.; Reinoso-Cobo, A.; Nieto-Gil, P.; Ortega-Avila, AB.; Gijon-Nogueron, G. A Systematic Review to Identify the Effects of Biologics in the Feet of Patients with Rheumatoid Arthritis. Medicina (Kaunas) 2020, 57, 23. doi: 10.3390/medicina57010023.
- Is necessary a sub-section of :… “3.3. Complications of surgery…”
Response: We followed the author guidelines of the journal. We believe that the “Complications of surgery” is an important subsection. The title for “3.2” was “Radiographic measurements.” Thus, we need a subsection of “3.3” for this study.
- Conclusion. Conclusion must respond to the main aim of the study. It is advisable that conclusion be clear and concise. Therefore, it is not recommended that its length be greater than some lines.
Response: Thank you for your valuable advice.
We have revised the Conclusions section as follows:
“Joint-preserving forefoot surgery involving the modified Mann method for the big toe and offset osteotomy for the lesser toes may prevent function loss after the restoration of forefoot deformities compared with resection arthroplasty for big and lesser toes in combination with biologic therapy for RA.”
- Limitations: broader studies are required.
Response: Thank you for pointing it out. We have added this in our manuscript as a suggestion for future studies.
“In the future, large randomized prospective studies with a multicenter approach and a longer follow-up duration should be conducted.”
References. I would like to congratulate you because 100% of the information you have referenced comes from articles published in scientific journals.
Congratulations on this wonderful article!! I think the authors have worked very closely considering methodology, outcomes. This will impact on international practice and help us reflect on the work we do. Thank you for this invitation to provide a review to evaluate this article for publication. I am delighted to have been invited to review this work.
Response: We are elated to have received such a positive feedback on your choice of references; this will encourage us further and keep us motivated. Thank you again for such wonderful and detailed comments and suggestions.
